# Improving and Streamlining Gene Editing in *Yarrowia lipolytica* via Integration of Engineered Cas9 Protein

**DOI:** 10.3390/jof10010063

**Published:** 2024-01-12

**Authors:** Baixi Zhang, Jiacan Cao

**Affiliations:** 1School of Food Science and Technology, Jiangnan University, Wuxi 214122, China; 6210113158@stu.jiangnan.edu.cn; 2National Engineering Research Center for Functional Food, Jiangnan University, Wuxi 214122, China

**Keywords:** *Yarrowia lipolytica*, genome editing, CRISPR/Cas9

## Abstract

The oleaginous yeast *Yarrowia lipolytica* is a prominent subject of biorefinery research due to its exceptional performance in oil production, exogenous protein secretion, and utilization of various inexpensive carbon sources. Many CRISPR/Cas9 genome-editing systems have been developed for *Y. lipolytica* to meet the high demand for metabolic engineering studies. However, these systems often necessitate an additional outgrowth step to achieve high gene editing efficiency. In this study, we introduced the eSpCas9 protein, derived from the *Streptococcus pyogenes* Cas9(SpCas9) protein, into the *Y. lipolytica* genome to enhance gene editing efficiency and fidelity, and subsequently explored the optimal expression level of *eSpCas9* gene by utilizing different promoters and selecting various growth periods for yeast transformation. The results demonstrated that the integrated *eSpCas9* gene editing system significantly enhanced gene editing efficiency, increasing from 16.61% to 86.09% on *TRP1* and from 33.61% to 95.19% on *LIP2*, all without the need for a time-consuming outgrowth step. Furthermore, growth curves and dilution assays indicated that the consistent expression of eSpCas9 protein slightly suppressed the growth of *Y. lipolytica*, revealing that strong inducible promoters may be a potential avenue for future research. This work simplifies the gene editing process in *Y. lipolytica*, thus advancing its potential as a natural product synthesis chassis and providing valuable insights for other comparable microorganisms.

## 1. Introduction

As a result of recent advances in gene editing technology, metabolic engineering, and synthetic biology, an increasing number of nonconventional microorganisms with unique characteristics have become hosts for synthesizing natural products [1,2,3]. Among these highly potential microorganisms, the oleaginous yeast *Yarrowia lipolytica* is a prominent subject of recent biorefinery research [4]. *Y. lipolytica* is an ideal host for industrial biomanufacturing, due to its excellent performance in oil production, exogenous protein secretion, and its ability to utilize various inexpensive carbon sources such as glycerol, alkanes, and acetic acid [5,6,7]. So far, *Y. lipolytica* has been used to produce various valuable products, including polyunsaturated fatty acids EPA [8] and CLA [9,10], terpenoid α-farnesene [11], and flavonoid naringenin [12], with several reaching commercialization. Despite numerous successful applications, there is still a lack of fundamental knowledge about *Y. lipolytica*. Only 44.5% of the 6448 coding genes are considered to be confidently annotated [13]. Thus, there is a need for more convenient and highly efficient gene editing tools to uncover the unknown secrets in *Y. lipolytica* and facilitate the construction of natural product synthesis chassis.

In recent years, the application of type II CRISPR/Cas9 systems from *Streptococcus pyogenes* has enhanced gene editing flexibility and efficiency in a wide range of organisms [14]. The CRISPR/Cas9 system comprises a Cas9 protein and a corresponding sgRNA. The sgRNA recognizes targeted sequences, while the Cas9 protein catalyzes a double-strand break (DSB) in the targeted genome loci. After the DSB forms, cells initiate the repair process, such as non-homologous end joining (NHEJ) and homologous recombination (HR) to repair the break loci, potentially introducing insertions and deletions (indels), or heterologous DNA fragments [15,16]. As a result, the CRISPR/Cas9 system has quickly gained widespread adoption due to its precision and high efficiency. Moreover, as metabolic engineering studies rapidly increase, various CRISPR/Cas9 systems have also been developed in *Y. lipolytica* as well [17].

Schwartz et al. [18] were the first to establish the CRISPR/Cas9 system in *Y. lipolytica*. Their focus was on identifying the best promoters for synthesis of sgRNA to enhance gene editing efficiency, resulting in high efficiency during single gene disruption by NHEJ. Subsequently, Gao et al. [19] developed a similar all-in-one CRISPR/Cas9 system, and successfully achieved double and triple gene disruption through NHEJ in *Y. lipolytica*. However, the CRISPR/Cas9 systems for *Y. lipolytica* mentioned above require an extended outgrowth step of 2-4 days following transformation to achieve high gene editing efficiency. While the outgrowth step has significantly increased gene editing efficiency by three to four times, it also presents some issues. Firstly, the gene editing process is extended by at least two days, making it more time-consuming. Besides, with the extension of cultivation time after transformation, the persistence of Cas9-sgRNA complex might also lead to increased frequencies of off-target mutations [20,21]. Moreover, the Cas9 nucleases of the aforementioned systems are delivered by plasmid vector with the size over 10kb, resulting in inconveniences when carrying out multiple genes’ editing [22].

In this study, we initially introduced an engineered Cas9 protein, eSpCas9 [23], to the CRISPR/Cas9 system of *Y. lipolytica*, to improve the fidelity and efficiency of gene editing. Subsequently, we integrated the eSpCas9 gene into the genome of *Y. lipolytica*, and investigated the optimal expression level of eSpCas9 by employing different promoters and selecting various growth periods for yeast transformation. The objective of this work is to propose a strategy for streamlining the gene editing workflow in *Y. lipolytica*, with the potential to expand the synthetic toolbox of *Y. lipolytica* and provide insights for other similar microorganisms.

## 2. Materials and Methods

### 2.1. Strains, Media, and Culture Conditions

*Escherichia coli* DH5α was used for plasmid construction and propagation. All *E. coli* cultures were cultured in Luria–Bertani (LB) medium at 37 °C. Suitable antibiotics (100 mg/L ampicillin or 50 mg/L kanamycin) were added to the LB medium when necessary. *Y. lipolytica* Po1f (Catherine Madzak, Paris-Saclay University, INRA, AgroParisTech, UMR SayFood, Palaiseau, France), a leucine and uracil auxotrophic strain, was used as the base strain in this study. All engineered strains are listed in Table 1. The *Y. lipolytica* cells stored in glycerol stock were initialized by streaking onto a YPD plate and cultivated at 28 °C overnight. A single colony from the plate was cultivated overnight in 5 mL YPD medium (1% yeast extract, 2% tryptone and 2% glucose) to create a seed culture, which was then transferred to culture flasks with a 1% inoculation dose for transformation. Post-transformation, the yeast cells were cultivated in appropriate selective media. YNBD medium (0.67% yeast nitrogen base (without amino acids) and 2% glucose) and YNBD-triglyceride medium (0.67% yeast nitrogen base (without amino acids) and 0.5% triglyceride) were used for *LIP2* disruption transformants. YNBD-leu medium (YNBD medium supplemented with 0.01% Leucine) was used for transformants with pINA1312 plasmid. YNBD-trp medium (YNBD supplemented with 0.01% tryptophan) was used for *TRP1* disruption transformants. Then, 2% agar was added to the solid medium. All *Y. lipolytica* culturing was conducted at 28 °C, with liquid cultures shaken at 200 rpm.

### 2.2. Plasmid Construction

Plasmids are listed in Table 1; primers are listed in Table 2. Representative plasmid maps can be found in Appendix A. The plasmid peCASyl, derived from pCASyl-trp, was synthesized by GenScript Biotech Co., Ltd. (Nanjing, China), which contained a codon-optimized *eSpCas9* cassette and an empty sgRNA cassette. The sequence of the codon-optimized *eSpCas9* gene is available in Appendix A. The plasmid pUC57-USA1B16 was derived from pUC57-USA1B8 [9]. The UAS1B8 fragment was obtained from pUC57-UAS1B8 by digestion with *Hind*III and *Xba*I restriction enzymes, and then ligated with pUC-UAS1B8 cut with *Hind*III and *Spe*I using T4 DNA Ligase (Vayzme, Nanjing, China), resulting in pUC57-USA1B16. All remaining plasmids assemblies were completed using a ClonExpress Ultra One Step Cloning Kit V2 (Vayzme, Nanjing, China). Plasmids pINA1312-TEFin-eCAS9 and pINA1312-UAS16TEFin-eCAS9 were derived from pINA1312. The *eSpCas9* cassette was amplified with primer pairs eCAS9-F10/eCAS9-R10 from peCASyl. The vector fragment from pINA1312 was amplified with primer pairs pINA1312-F/pINA1312-R for pINA1312-TEFin-eCAS9, and primer pairs pINA1312-F/pINA1312-R-2 for pINA1312-UAS16TEFin-eCAS9. The pUC57-USA1B16 plasmid was digested by restrictive endonucleases *Spe I* and *Xba I* to obtain 16 tandem copies of UAS1B for construction of pINA1312-UAS16TEFin-eCAS9. Plasmid pCEN1-Leu-gRNA was derived from peCASyl, which abandoned the *eSpCas9* cassette. Primer pairs Mig1t-F2/TEF-R2 and AmpR-F/ORI001-R were used to amplified the required fragments for construction of pCEN1-Leu-gRNA. As for the insertion of 20 bp gRNA in peCASyl-trp, pCEN1-Leu-gTrp, and pCEN1-Leu-gLip, primer pairs gRNA-Trp-F/gRNA-Trp-R were used for gRNA of *TRP1* (5′-CGATGGCGTCCTGATCCAGT-3′), and primer pairs gRNA-LIP-F/gRNA-LIP-R were used for gRNA of *TRP1* (5′-CAGTTGAAGGGCTTGAAGAT-3′). All products from PCR or restrictive endonuclease were recovered by gel extraction and electrophoresis using The Fast Pure Gel DNA Extraction Mini Kit (Vayzme, Nanjing, China). The recombination products were transformed into *E. coli* DH5α, and cultured at 37 °C overnight. Single colonies were then picked for final verification. Plasmid DNA was extracted using TIANprep Mini Plasmid Kit from Tiangen Biotech Co., Ltd. (Beijing, China).

### 2.3. Yeast Transformation

The transformation of *Y. lipolytica* utilized the Frozen-EZ Yeast Transformation II Kit (Zymo Research, Orange, CA, USA). The number of cells was adjusted to a range of 1 × 10^7^ to 1 × 10^8^ before transforming. Subsequently, 1 mL of YPD medium was introduced into the transformation mixture, followed by cultivation at 28 °C and 200 rpm for 2 h to facilitate cell recovery. Post-recovery, the cells were harvested via centrifugation at 13,400× *g* for 1 min and then plated onto suitable medium.

### 2.4. Phenotype Verification

The phenotype verification involved transferring single transformant colonies from the original plate to two plates. One plate served for selecting transformants with distinct phenotypes, while the other functioned as a non-selective control. YNBD and YNBD-trp media were utilized to confirm the *TRP1* disruption transformants, while YNBD and YNBD-triglyceride media were used to validate the *LIP2* disruption transformants.

### 2.5. Colony PCR Confirmation and Sequencing

To obtain the genome template, the single colony was dispersed in 20 μL of a 0.25% SDS solution, followed by incubation at 90 °C for 3 min. The PCR was performed using the 2× Phanta Max Master Mix (Dye Plus) from Vazyme Biotech Co., Ltd. (Nanjing, China). And the resulting products were verified by DNA electrophoresis or sequenced as necessary by Sangon Biotech Co., Ltd. (Shanghai, China). *eSpCas9* was confirmed using primer pairs eCAS9-F/eCAS9-R; *TRP1* was confirmed using primer pairs TRP-F/TRP-R; and *LIP2* was confirmed using primer pairs LIP-F/LIP-R.

### 2.6. Real-Time Quantitative PCR

Total RNA from the yeast was extracted and purified using the RNAprep Pure Plant Kit from Tiangen Biotech Co., Ltd. (Beijing, China). Subsequently, the cDNA was synthesized using the HiScript III 1st cDNA synthesis kit from Vazyme Biotech Co., Ltd. (Nanjing, China). For the qPCR reaction system, SYBR GREEN 2×PCR Master Mix (ABI, Carlsbad, CA, USA) was employed. In the relative copy number measurement, strain Po1g was utilized as the control organism with a single copy of both the *URA3* and *SUC2* target sequences. The primer pairs targeting *URA3* are ura3-F/ura3-R, and the primer pairs targeting *SUC2* are Suc2-F/Suc2-R. As for the relative expression level measurement, *ACT1* (YALI0D08272g) was employed as the reference gene due to its relatively constant expression. The primer pairs targeting *eSpCas9* are qeCAS9-F2/qeCAS9-R2, and the primer pairs targeting *ACT1* are ACT-F/ACT-R. Gene expression changes were assessed using the 2^−ΔΔCt^ method, and all samples were prepared in triplicate to obtain the Ct value.

## 3. Results and Discussion

### 3.1. Gene Editing Efficiency of eSpCas9 in Y. lipolytica

Off-target effects arise when the nuclease activity of Cas9 is triggered at sites where the RNA guide sequence exhibits imperfect complementarity with off-target genomic sites, presenting a significant challenge for genome editing applications [26]. To improve specificity and efficiency in gene editing, several variants of SpCas9 were initially created through mutagenesis of protein regions involved in target DNA binding [27]. One such variant, eSpCas9, was engineered with K848A, K1003A, and R1060A mutations using alanine mutagenesis of positively charged residues in the non-target strand binding groove. This was to reduce affinity and promote rehybridization between the target and non-target DNA strands, resulting in highly efficient and more specific gene editing in mammalian cells compared to wild-type cells [23]. Building upon the CRISPR/Cas9 system established by Gao [19], we developed all-in-one CRISPR plasmids, termed peCASyl, containing a de novo codon-optimized version of the eSpCas9 gene, with the aim of achieving higher gene editing efficiency in *Y. lipolytica*.

To assess the efficacy of peCASyl in gene editing, we targeted the *TRP1* gene for editing, as its absence renders strains unable to grow in tryptophan-deficient media (Figure 1A). As the results demonstrate, a disruption efficiency of 35.5% was observed in strains transformed with peCASyl, and the disruption efficiency further increased to 71.23% and 77.21% after outgrowth periods of 2 and 4 days, respectively (Figure 2). It is worth noting that the introduction of eSpCas9 significantly enhances the efficiency of gene editing compared to the original CRISPR/Cas9 system, whether engaging in an additional outgrowth step or not (Figure 2). The efficient expression of the recombinant protein, facilitated by codon optimization and minimal off-target effects, likely contributes to this improvement in gene editing efficiency. Furthermore, there appears to be no substantial difference in gene editing efficiency for peCASyl between 2 and 4 days of outgrowth analyzed with an unpaired Student’s *t*-test (Figure 2), suggesting the potential to shorten the gene editing workflow while maintaining high efficiency. These findings indicate that the introduction of *eSpCas9* into *Y. lipolytica* enables normal function and enhances gene editing efficiencies, akin to its performance in mammalian cells. The use of *eSpCas9* is particularly suitable for specific experimental requirements or for expanding the capabilities and applications of CRISPR toolboxes, such as single-nucleotide conversion, which may provide potential insights into the synthetic toolbox of *Y. lipolytica* [28].

### 3.2. Integration of eSpCas9 in Y. lipolytica

As previously mentioned, most CRISPR/Cas9 systems in *Y. lipolytica* use plasmids to deliver Cas9 into cells, often necessitating additional outgrowth to achieve high gene editing efficiency, and potentially resulting in oversized plasmids. Therefore, we opted to develop a recombinant yeast strain with integrated *eSpCas9* for more efficient and convenient gene editing [29]. Taking into account the subsequent marker rescue, we selected the single-copy vector pINA1312, which contains the non-defective *ura3d1* gene for mono-copy expression, to integrate the *eSpCas9* gene into *Y. lipolytica* [25]. Initially, we constructed the plasmid pINA1312-TEFin-eCas9, which inserted an *eSpCas9* cassette into vector pINA1312. Subsequently, we linearized the plasmid with restrictive endonucleases *Not*I to obtain a DNA fragment containing ura3d1 marker, *eSpCas9* cassette and Zeta sequence, which was subsequently introduced into yeast genome through transformation. The agarose gel electrophoresis results confirmed the successful integration of the *eSpCas9* gene into the yeast strains, designated as po1fe-3 (Figure 3A). Following the integration of the expression cassette, we confirmed the copy numbers using real-time quantitative PCR. *Y. lipolytica* Po1g, which contains a single copy of the *URA3* and *SUC2* target sequences, was employed as a control organism. Given the co-existence of *URA3* and *eSpCas9* within the expression cassette, their relative copy numbers were assumed to be equivalent. As depicted in Figure 3B, all transformants exhibited less than two copies.

To assess the functionality of integrated *eSpCas9*, we introduced a replicative plasmid containing the 20bp gRNA of the *TRP1* gene, named pCEN1-Leu-gTrp, which abandoned the *eSpCas9* cassette on peCASyl. As depicted in Table 3, the disruption efficiency achieved with the system po1fe3-1/pCEN1-Leu-gTrp reached 86.09% ± 3.28 on *TRP1* without the need for outgrowth, compared to a disruption efficiency of 35.50% ± 5.22 for po1f/peCASyl-trp and 16.61% ± 5.31 for po1f/pCASyl-trp without outgrowth as well. As anticipated, the heterologous expression of *eSpCas9* in *Y. lipolytica* significantly enhances gene editing efficiency. This is due to the pre-existing abundance of Cas9 protein in the cell before the delivery and transcription of the sgRNA cassette. This abundance immediately facilitates the formation of Cas9-sgRNA complexes within the cell, ultimately leading to a higher rate of double-strand break formation at the target site. Consequently, we achieved a five-fold increase in gene editing efficiency compared to the original CRISPR/Cas9 system developed by Gao [19], without the need for outgrowth. This implies that we can forego the time-consuming outgrowth step and reduce the risk of off-target effects caused by the continuous presence of Cas9-sgRNA complexes through use of the high-fidelity eSpCas9 protein [23,27].

It is widely recognized that an increase in gene editing efficiency typically leads to a decrease in the survival rate of edited cells, as more cells are susceptible to genomic damage [19]. As demonstrated in Table 3, we attained comparable transformation efficiency between the po1fe3-1 and peCASyl systems while significantly enhancing gene editing efficiency. We attribute this difference to the plasmid size, as the po1fe3-1 system requires a smaller plasmid than the peCASyl system, which does not require transfection of the *eSpCas9* gene. This smaller plasmid size allows more cells to be transfected with the plasmid under the same mass of plasmid, yielding more transformants.

Additionally, to further validate the po1fe strains, we expanded the testing gene to include the *LIP2* gene. The *LIP2* gene encodes extracellular lipase, and in its absence, the strains exhibit a reduced halo of triglyceride hydrolysis when cultured on medium containing tributyrin (Figure 1B) [30]. The results show a substantial enhancement in gene editing efficiency for the *LIP2* gene, increasing from 33.61% ± 7.08 for the pCASyl system to 95.19% ± 1.7 for the po1fe3-1 system (Table 3). The trend of transformation efficiency among the three systems is consistent with that of the *TRP1* gene. Additionally, the transformation efficiency in *LIP2* exhibited a specific reduction compared to *TRP1*, which can be attributed to the higher gene editing efficiency in *LIP2*, making more cells vulnerable to genomic damage.

### 3.3. Effect of Enhancing Gene Expression Level of Integrated eSpCas9 on Gene Editing Efficiency

Previous studies have shown that increasing the expression of Cas9 nuclease can enhance the indel rate in mammalian cells [31]. While we achieved high gene editing efficiency with po1fe3-1, there is potential for further improvement by regulating the *eSpCas9* expression level. Therefore, we enhanced the original promoter TEFin by adding 16 tandem copies of an upstream activation sequence (UAS) from *P_TEF_* [32,33]. Subsequently, we integrated the eSpCas9 expression cassette containing the enhanced promoter into yeast cells to create a recombinant strain named po1fe4 in the same way as po1fe3, choosing po1fe4-2 as the tested strain based on relative copy number data. Additionally, according to the experience in previous studies, the expression level of heterologous proteins in *Y. lipolytica* varies with the growth phases [9]. This suggests that the growth phases may potentially influence the intracellular eSpCas9 protein content before transforming the plasmids containing the sgRNA cassette, which could consequently impact gene editing efficiency. Therefore, to confirm this hypothesis, we tested the gene editing efficiency of the selected genes after enhancing the promoter and preparing receptive cells at different growth cycles.

At first, we assessed the relative expression levels among the above strains using real-time quantitative PCR. As shown in Figure 4A, the enhanced promoter led to a nearly 12-fold increase in the eSpCas9 expression level. While the influence of the growth cycle on eSpCas9 expression was not as significant as the influence of the enhanced promoter, the most significant difference in eSpCas9 expression between the early log-phase and mid-stationary phase was less than two-fold (Figure 4B). However, as depicted in Figure 5, whether the *eSpCas9* expression level was significantly increased by enhancing the promoter or slightly adjusted by transforming at different growth phases, there were no significant variances in gene editing efficiency of *TRP1* analyzed with an unpaired Student’s t-test, showing an efficiency of approximately 90%. We speculated that the expression of *eSpCas9* reached a sufficient level to achieve the highest editing efficiency under the regulation of promoter TEFin, indicating no need to pursue a higher expression level of the nuclease. Furthermore, the limitation of sgRNA expression may also be the reason for the lack of further improvement in editing efficiency, as the level of the sgRNA-Cas9 complex failed to increase with the level of eSpCas9 nuclease [34].

### 3.4. Effect of Integrated eSpCas9 on Growth of Y. lipolytica

Cas9 nucleases are recognized for their toxicity to various species, especially when expressed at high levels in host cells [35]. Therefore, we investigated the impact of *eSpCas9* on the growth of *Y. lipolytica* at varying expression levels using growth curves. Initially, we transformed pINA1312 plasmids into po1f to restore the *ura* auxotroph, constructing po1f-ura+ as a control strain. Subsequently, we prepared seed cultures of the strains po1f-ura+, po1fe3-1, po1fe4-2, and inoculated them into fresh YPD medium for cultivation at 28 °C, 200 rpm, measuring the OD_600_ every 4 h as the data of growth curves. As depicted in Figure 6A, the growth of po1fe3-1 closely resembled that of the control strain po1f-ura+. Despite significantly increased expression levels using a UAS16-TEFin promoter, the growth of po1fe4-2 only suffered slight suppression (Figure 6A). In order to further explore the effect of Cas9 on strains’ growth, a dilution assay was performed [36]. The cultures of po1f-ura+, po1fe3-1 and po1fe4-2 were adjusted to an OD_600_ of 2.5, followed by serial dilutions, and then plating of 5 μL spots on YNBD-Leu plates. As shown in Figure 6B, after cultivation for 12 h and 24 h, the growth of strains integrated with *eSpCas9*, po1fe3-1 and po1fe4-2, exhibited slightly impaired growth compared to the control strain po1f-ura+. However, similar delayed growth between po1fe3-1 and po1fe4-2 was not readily observed, indicating a high tolerance of *Y. lipolytica* to heterologous protein to some extent. In summary, control of eSpCas9 expression using strong inducible promoters may be a potential avenue for future research.

## 4. Conclusions

In this study, we developed an enhanced gene editing system for *Y. lipolytica* based on the CRISPR/Cas9 system. The high-fidelity eSpCas9 protein was incorporated into the system to improve gene editing efficiency. Subsequently, *eSpCas9* was integrated into the genome of *Y. lipolytica* under the control of the TEFin promoter, resulting in significantly improved gene editing efficiency without the need for outgrowth. To further maximize efficiency, the TEFin promoter was replaced by the stronger UAS16-TEFin promoter. However, the potential improvement in gene editing was constrained by the original high efficiency and expression level of sgRNA. Additionally, we confirmed that the consistent expression of eSpCas9 protein slightly suppressed the growth of *Y. lipolytica*, revealing that the strong inducible promoters may be a favorable approach for future research. This approach simplifies the gene editing process in *Y. lipolytica*, which has the potential to broaden the synthetic capabilities of *Y. lipolytica* and offer valuable insights for other comparable microorganisms.

## Figures and Tables

**Figure 1 jof-10-00063-f001:**
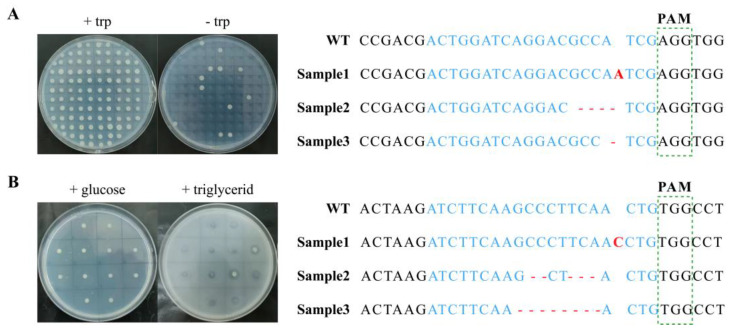
Phenotype and sequence confirmation of *TRP1* and *LIP2* disruption strains. (**A**) Verification of *TRP1* disruption strains. Transformants that failed to grow on YNBD were *trp*− mutants. A representative sample is shown for the alignments of the *TRP1* gene sequence from selected *trp*− mutants. (**B**) Verification of *LIP2* disruption strains. Transformants that exhibited a reduced halo of triglyceride hydrolysis on YNBD-triglyceride were *lip*− mutans. A representative sample is shown for the alignments of the *LIP2* gene sequence from selected *lip*− mutants. The blue represents gRNA sequence, the red represents indels, and the sequence in green rectangle represents PAM sequence.

**Figure 2 jof-10-00063-f002:**
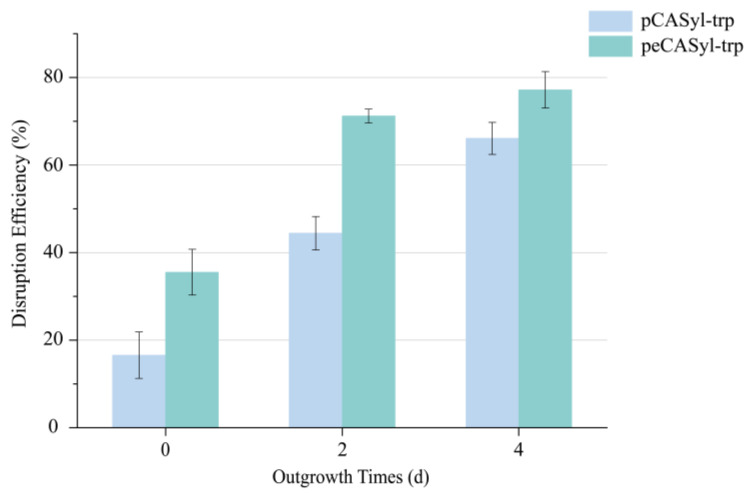
The disruption efficiency of pCASyl-trp and peCASyl-trp on gene *TRP1*. The values and error bars represent the average readings and standard deviations for three independent experiments.

**Figure 3 jof-10-00063-f003:**
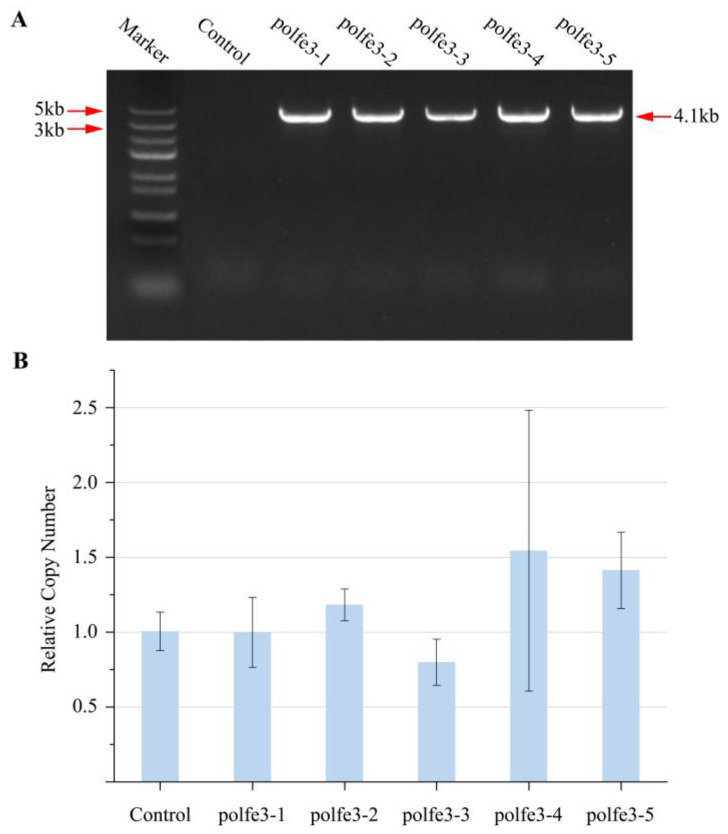
Confirmation of *eSpCas9* integration. (**A**) Agarose gel electrophoresis of an *eSpCAS9* fragment amplified with primer pairs eCas9-F/eCas9-R. (**B**) Relative copy number of *eSpCAS9* in transformants via real time PCR; *Y. lipolytica* Po1g containing a single copy of both the *URA3* and *SUC2* was set as a control. The values and error bars represent the average readings and standard deviations for three independent experiments.

**Figure 4 jof-10-00063-f004:**
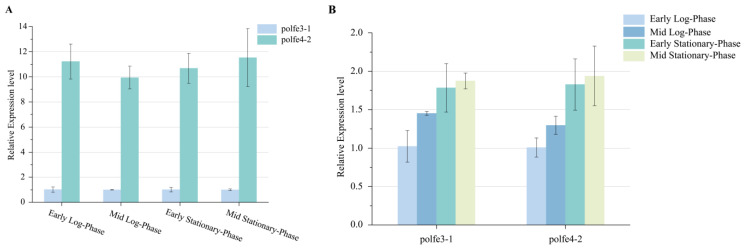
The relative expression level of eSpCas9 in po1fe3-1 and po1fe4-2. (**A**) Relative expression level with po1fe 3-1 as control. (**B**) Relative expression level with early log phase as control. Values and error bars represent the average readings and standard deviations for three independent experiments.

**Figure 5 jof-10-00063-f005:**
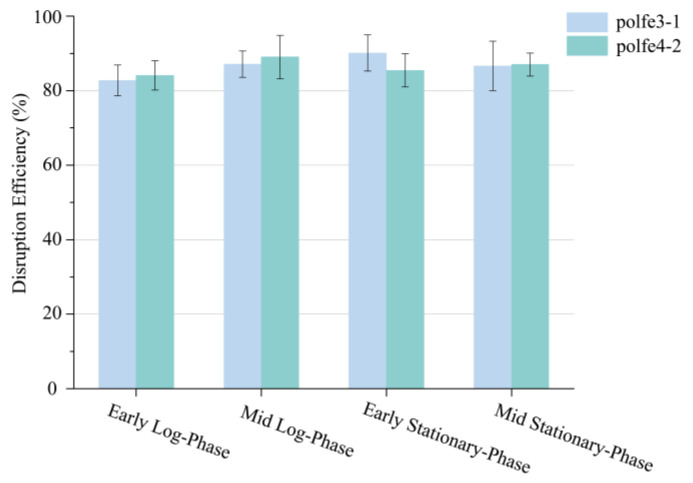
The disruption efficiency of *TRP1* under different *eSpCas9* expression levels. po1fe3-1 and po1fe4-2 shows that *eSpCas9* expression was controlled under promoter *pTEFin* and *pUAS16TEFin*; the *x* axis shows the preparation of receptive cells in different growth cycles. Values and error bars represent the average readings and standard deviations for three independent experiments.

**Figure 6 jof-10-00063-f006:**
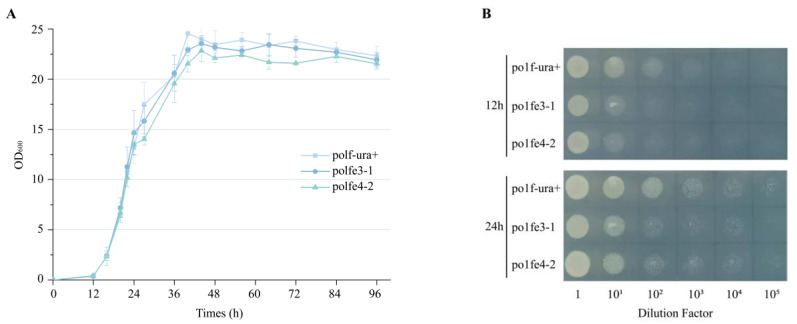
The effect of eSpCas9 protein on growth of *Y. lipolytica*. Strain po1f-ura+ was set as control. (**A**) Growth curves of po1f-ura+, po1fe3-1, po1fe4-2. Values and error bars represent the average readings and standard deviations for three independent experiments. (**B**) Dilution assay of po1f-ura+, po1fe3-1, po1fe4-2, with a consistent initial OD_600_ of 2.5.

**Table 1 jof-10-00063-t001:** Strains and plasmids used in this work.

Strains or Plasmids	Characteristics	Source or References
*Strains*		
Po1f	*MatA*, *leu2-270*, *ura3-302*, *xpr2-322*, *axp-2*, *Leu−*, *Ura−*, Δ*AEP*, Δ*AXP*, *Suc+*	[24]
Po1g	*MatA*, *leu2-270*, *ura3-302: :URA3*, *xpr2-322*, *axp-2*, *Leu−*, Δ*AEP*, Δ*AXP*, *Suc+*, *pBR*	[24]
Po1fe-3	Po1f, pINA1312-TEFin-eCAS9 (*URA3*, *eSpCas9*)	This work
Po1fe-4	Po1f, pINA1312-UAS16TEFin-eCAS9 (*URA3*, *eSpCas9*)	This work
Po1f-ura+	Po1f, pINA1312 (*URA3*)	This work
*Plasmids*		
pCASyl-trp	*Y. lipolytica*-replicative plasmid, containing *TRP1* Guide RNA module and Cas9 expression cassette	[19]
pINA1312	*Y. lipolytica*-integrative plasmid, *Kan*, *hp4d*, *XPR2t*, *ura3d1*	[25]
pUC57-UAS1B8	Containing 8 tandem copies of UAS1B flanked with *Cla*I-*Hind*III-*Spe*I siteson the 5′-end and *Mlu*I-*Xba*I sites on the 3′-end.	[9]
pUC57-USA1B16	Derived from pUC57-UAS1B8, containing 16 tandem copies of UAS1B	This work
peCASyl	Derived from pCASyl-trp, replacing Cas9 gene to codon-optimized *eSpCas9* gene	This work
peCASyl-trp	Derived from peCASyl, containing *TRP1* Guide RNA module	This work
pINA1312-TEFin-eCAS9	pINA1312 containing TEFin-eSpCas9 cassette	This work
pINA1312-UAS16TEFin-eCAS9	pINA1312 containing UAS16TEFin-eSpCas9 cassette	This work
pCEN1-Leu-gRNA	Derived from pCASyl-trp, containing empty sgRNA cassette	This work
pCEN1-Leu-gTRP1	Derived from pCEN1-Leu-gRNA, containing *TRP1* Guide RNA module	This work
pCEN1-Leu-gLIP2	Derived from pCEN1-Leu-gRNA, containing *LIP2* Guide RNA module	This work

**Table 2 jof-10-00063-t002:** Primers used in this work.

Primers	Sequence (5′-3′)
eCAS9-F10	ACATGGAATTCGGACACGGGccgacgcgtctgtacaga
eCAS9-R10	AGTCTGCAGCCCAAGCTAGCgcccttttgggtttgtcgac
pINA1312-F	CCCGTGTCCGAATTCCATGTGTAAC
pINA1312-R	GCTAGCTTGGGCTGCAGACTAAATT
pINA1312-R-2	GACACCTCAGCATGCACTAGgctagcttgggctgcagactaaatt
Mig1t-F2	agacatagcggccgcttcgaaaacccaaaagggccgaagg
TEF-R2	ggggttccgcacacatttccagagaccgggttggcgg
AmpR-F	ggaaatgtgtgcggaacccc
ORI001-R	tcgaagcggccgctatgtct
NdeI-F	GTGCGGTATTTCACACCGCAtatggtgcactctcagtacaatctgc
NdeI-R	tgcggtgtgaaataccgcac
gRNA-Trp-F	CGATGGCGTCCTGATCCAGTgacgagcttactcgtttcg
gRNA-Trp-R	ACTGGATCAGGACGCCATCGgttttagagctagaaatagcaagt
gRNA-LIP-F	CAGTTGAAGGGCTTGAAGATgacgagcttactcgtttcg
gRNA-LIP-R	ATCTTCAAGCCCTTCAACTGgttttagagctagaaatagcaagt
eCas9-F	ATCGCCACCGAGCTGACT
eCas9-R	GACAAGAAGTACTCAATCGGCCTGG
TRP-F	ATGGACTTTCTCTACTCTTCGACAT
TRP-R	TTACCCCCTGGCGTTTTTGAC
LIP-F	CCCAAGTACCAGCTCCCCTC
LIP-R	CCACAGACACCCTCGGTGAC
Suc2-F	CTACGGTTCAGCATTAGGTATTG
Suc2-R	GACCAGGGACCAGCATTAC
ura3-F	GTGCTTCTCTGGATGTTACC
ura3-R	CAATATCTGCGAACTTTCTGTC
ACT2-F	TCCAGGCCGTCCTCTCCC
ACT2-R	GGCCAGCCATATCGAGTCGCA
qeCAS9-F2	TAAGCAACCGTAGGGGAATC
qeCAS9-R2	GTGTGGGACAAGGGCAGAGA

**Table 3 jof-10-00063-t003:** The disruption efficiency and transformation efficiency of different gene editing systems targeting *TRP1* and *LIP2* without outgrowth.

Targeted Gene	Strain/Palsmid	Disruption Efficiency (%)	Transformation Efficiency (CFU/ng)
*TRP1*	po1f/pCASyl-trp	16.61 ± 5.31	689.67 ± 62.85
	po1f/peCASyl-trp	35.50 ± 5.22	536.67 ± 38.52
	po1fe3-1/pCEN1-Leu-gTrp	86.09 ± 3.28	626.00 ± 57.20
*LIP2*	po1f/pCASyl-trp	33.61 ± 7.08	281.67 ± 57.62
	po1f/peCASyl-trp	62.78 ± 5.20	211.33 ± 69.94
	po1fe3-1/pCEN1-Leu-gTrp	95.19 ± 1.70	189.00 ± 50.04

Note: Results are presented as mean ± SD for three independent experiments.

## Data Availability

The data supporting the conclusions of this study are available and included within the article.

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
