# Peer review of "Improving and Streamlining Gene Editing in Yarrowia lipolytica via Integration of Engineered Cas9 Protein"

_jof, 2024, doi:10.3390/jof10010063_

Round 1
Reviewer 1 Report
Comments and Suggestions for Authors
In this manuscript, Zhang and Cao present an improved CRISPR/Cas9-based gene editing method for the oleaginous yeast Yarrowia lipolytica. By introducing an engineered Cas9 protein as single copy the authors accomplish a higher gene editing efficiency in a shorter time when compared with the already available methods. Therefore, this methodology would be of interest for Yarrowia researchers. Please address the following issues prior publication.
Line 15:
Define “eSpCas9”.
Fig 1: labels
Is “YNBD-trp” indicating lack or addition of tryptophan?
In any case, “YNB + Trp” and “YNBD + tryglicerides” would be clearer. On the other hand, it looks like labels are swapped in panel B.
Lines 193-195:
Please explain that selection for ura+ prototrophs was employed.
Fig. 3B:
The relative copy number in control should be around 0 as no eSpCas9 was integrated.
In addition, please assess URA3 as control of single copy integration.
Table 3:
Is the data shown obtained after 2 days? Please state this accordingly.
Fig. 5:
It is unclear what the authors refer to. I would understand that gene editing efficiency was assessed at different stages of the cell cycle. However, I cannot see the point of that. Please clarify.
Fig. 6:
Please perform dilution assays to reinforce the point that high eSpCas9 expression is not toxic.
Comments on the Quality of English LanguageTable 1:
Typo “replacative”.
Reviewer 2 Report
Comments and Suggestions for Authors
In this paper, the authors report a significant improvement in genome editing efficiency of Y. lypolytica by using modified Cas9.Furthermore, they show that genome-integrated Cas9 expression is extremely more efficient for genome editing than Cas9 expressed in a plasmid.
Although the improvement of eSpCas9 expression is also examined, it is shown that sufficient genome editing efficiency can be achieved with the constitutive promoter initially used.
This study is of great value as it will accelerate the molecular breeding of Y. lypolytica.
I point out only minor points.
1. The name of the plasmid is confusing. My understanding is as follows
pCASyl-trp: Cas9 and TRP1 guide RNA expression.
peCASyl: modified Cas9 and guide RNA expression.
peCASyl-trp: TRP1 guide RNA expression.
The reader may take pCASyl-trp and peCASyl-trp as similar. (I did, too, at first sight.)
2. line 195, the name of the organism should be in italics.
3. line 205, “transformed" should be “introduced” or "po1fe3-1 was transformed by"
4. I look forward to further development of author's work, such as control of Cas9 expression using tuneable or inducible promoters.
Reviewer 3 Report
Comments and Suggestions for Authors
In previous decades most of research efforts were concentrated on studying of several model objects. One of them is Saccharomyces cerevisiae yeast. Other yeast species were usually referred to as “non-conventional”. But development of new methods and techniques, such as genome editing and “omics” technologies now allows performing comprehensive research of such species. It is very inspiring because such research allows comparing different yeast species on systemic level. This will lead to better understanding of evolutionary processes, role of yeast species in natural communities, etc. Active research of such “non-conventional” yeasts is also of big practical importance due to their use in biotechnology.
Reviewed study presents an approach that simplifies the gene editing process in Yarrowia lipolytica. This yeast is a perspective host that is used in biotechnology for oil and recombinant protein production. Thus such work definitely has practical importance.
36 “terpenoids α-farnesene” – maybe better “terpenoid α-farnesene”
37 “flavonoids naringenin” – maybe better “flavonoid naringenin”
55 “identifying the best promoters of sgRNA” – maybe better “identifying the best promoters for synthesis of sgRNA”
65-67 “Moreover, the Cas9 nucleases of the aforementioned systems are delivered by vector which generally cause the size of plasmids over 10kb, resulting in inconveniences when operating multiple genes editing.” – maybe better to rephrase. For example: “Moreover, the Cas9 nucleases of the aforementioned systems are delivered by plasmid vector with the size over 10kb, resulting in inconveniences when operating multiple genes editing.” Or something like that.
83 “…, and all engineered strains are listed in Table 1.” – maybe it is better to make it a separate sentence: “All engineered strains are listed in Table 1.”
Table 1 - “replacative”
83 “The Y. lipolytica cells stored in glycerol stock was initialized…”
97-113 The objective of this work is stated as (72) “to propose a strategy for streamlining the gene editing workflow in Y. lipolytica”. Thus, comprehensible and reproducible description of plasmids and vectors that were generated and used in this work is required. It is not enough to describe construction of single pINA1312-UAS16TEFin-eCAS9 plasmid. Construction procedure for all plasmids should be described and their corresponding maps should be provided. Maybe not in the main text, but in supplementary materials. Codon optimized sequence of eSpCas9 gene should be also provided.
122 “Phenotype verification and sequencing” – there is nothing about sequencing
146, 188 “Y lipolytica” – “Y. lipolytica”
192 “integrated with eSpCas9” maybe “with integrated eSpCas9 gene”
195 “Y. lipolytica” should be italic.
No information on statistical analysis of the results is presented in the paper. For example in line 177 it is stated “Furthermore, there appears to be no substantial difference in gene editing efficiency for peCASyl between 2 and 4 days of outgrowth.” It should be mentioned which statistical criteria was used to compare results of the experiments and how many replications were analyzed. In Figure 2 and other figures it should be mentioned what is represented by error bars. Starting from line 207 “86.09 % ± 3.28”, further on in the text and in Table 3 it should be also described, how many replicas were analyzed and what is presented as errors (SEM or SD?).
194 Was pINA1312 vector linearized before transformation? Where in the Y. lipolytica genome it was integrated?
What primers were used for PCR analysis presented in Figure 3A?
Experiments in lines 205-218 are described very poorly. “To assess the functionality of integrated eSpCas9, we transformed a plasmid containing the 20bp gRNA of the TRP1 gene, named peCASyl-trp.” Which strains were transformed and with which plasmids? In Table 3 and in text results for pCASyl system are also analyzed. So pCASyl derived plasmid for TRP1 was also used?
Was po1fe3-1 strain transformed with peCASyl-trp plasmid in this experiment? As it is stated in Table 1 peCASyl-trp plasmid is based on peCASyl which contains codon-optimized eSpCas9 gene. So were there two expression cassettes for eSpCas9 gene in the resulting strain – one integrated in the genome and another on peCASyl-trp plasmid? Transformation of po1fe3-1 strain with peCASyl-trp levels out the idea of making vectors for gene editing in Y. lipolytica more compact and easy to handle. In my opinion a more suitable experiment would be to transform po1fe3-1 strain with a plasmid containing only sgRNA cassettes without eSpCas9 gene and analyze the efficiency of resulting system.
Unfortunately I was not able to open archive with original images blots, although I tried to do it on two different computers with two different programs.
Round 2
Reviewer 1 Report
Comments and Suggestions for Authors
Authors have addressed all my concerns. I am very satisfied with authors' response.
Reviewer 3 Report
Comments and Suggestions for Authors
I accept corrections and answers to questions.